# Natural Phenolic Compounds as Modifiers for Epoxidized Natural Rubber/Silica Hybrids

**DOI:** 10.3390/molecules27072214

**Published:** 2022-03-29

**Authors:** Olga Olejnik, Anna Masek

**Affiliations:** Faculty of Chemistry, Institute of Polymer and Dye Technology, Lodz University of Technology, Stefanowskiego 16, 90-537 Lodz, Poland; olga.olejnik@p.lodz.pl

**Keywords:** quercetin, epoxidized natural rubber, silica, gallic acid, tannic acid, biocomposite

## Abstract

Silica is a popular filler, but in epoxidized natural rubber, can act as a cross-linking agent. Unfortunately, a high amount of silica is necessary to obtain satisfactory tensile strength. Moreover, a high amount of silica in ENR/silica hybrids is associated with low elongation at break. In our paper, we propose natural phenolic compounds, including quercetin, tannic acid, and gallic acid as natural and safe additional crosslinkers dedicated to ENR/silica hybrids to obtain bio-elastomers with improved mechanical properties. Therefore, toxic crosslinkers, such as peroxides or harmful accelerators can be eliminated. The impact of selected natural phenolic compounds on crosslinking effect, mechanical properties, color, and chemical structure of ENR/silica composite have been analyzed. The obtained results indicated that only 3 phr of selected natural phenolic compounds is able to improve crosslinking effect as well as mechanical properties of ENR/silica hybrids. Moreover, some of the prepared materials tend to regain mechanical properties after reprocessing. Such materials containing only natural and safe ingredients have a chance of becoming novel elastomeric biomaterials dedicated to biomedical applications.

## 1. Introduction

The designing and preparation of pro-ecological composites, including bio-based elastomers, is still of great interest [1,2,3]. Natural Rubber (NR) derived from *Hevea brasiliensis* plant is a popular biopolymer, widely utilized to obtain such elastomeric biocomposites [4,5]. Nevertheless, the structure of natural rubber requires using traditional curing agents, including sulphur, activators, and organic accelerators, which are not environmentally friendly substances. The controlled chemical modification of natural rubber is helpful for creating materials with more reactive moieties. The most popular modified natural rubber is epoxidized natural rubber (ENR), which contains highly reactive oxirane rings. Such a material can be crosslinked using more pro-ecological substances and sometimes might be reprocessable [6]. Maleic anhydrate, carboxylic acids, or lignin are only a few examples of substances that can be chosen as crosslinking agents to prepare ENR-based environmentally friendly elastomeric biocomposites. Moreover, blending epoxidized natural rubber with biodegradable thermoplastics, such as polylactide or polycaprolactone, yields thermoplastic bio-elastomers [7,8]. Interestingly, silica, which is a popular active filler, is also able to react with oxirane groups of ENR chain and form crosslinks without any extra additives [9].

Such ENR/silica-based composites can be prepared in a simple way and save the pro-ecological character of epoxidized natural rubber. However, such elastomers require a high amount of filler to gain satisfying mechanical properties, including high tensile strength, which is also related to low elongation at break due to the stiffening effect [9]. Therefore, additional compounds, which are able to modify such composites to obtain better mechanical properties, are desired. It is also important to choose pro-ecological substances to maintain the environmentally friendly character of composites. In our research, we decided to choose natural phenolic compounds, including the most popular ones, such as quercetin, gallic acid, and tannic acid, which can modify ENR/silica hybrids to obtain eco-friendly composites with improved tensile strength and elongation at break. The safety of the selected substances, as well as epoxidized natural rubber, allows to prepare materials dedicated to biomedical engineering.

In comparison to gallic acid and tannic acid used in our research, quercetin is a typical representative of polyphenols, a type of flavonoids widely occurring in plants, including onion, tea, and apples [10]. These compounds reveal a strong antibacterial and antioxidant activity [11]. The anticancer potential of quercetin has also been discussed [12]. This polyphenol has five hydroxyl groups in the structure, which can be responsible not only for the stabilization effect [13] but also for the crosslinking effect [14,15]. Such an effect has been proved in the case of polymer technology, where epoxidized soybean oil-based thermosets were prepared [14], as well as in dental applications, where this flavonoid served as a primer in dentin bonding [10]. This compound was also used in other medical implementations, such as drug delivery [11]; therefore, quercetin seems to be an ideal component to create biocomposites safe for human beings.

Gallic acid (3,4,5-trihydroxybenzoic acid) presented in Figure 1 is a phenolic acid of plant origin, which can be extracted from numerous fruits and vegetables, for instance, grapes and tomatoes. Due to its low toxicity, natural origin as well as stabilizing properties, including antioxidant and antifungal effectiveness, gallic acid is a noteworthy compound, which can be dedicated for eco-friendly composites [16]. It has been proved that phenolic acids can act as cross-linkers for biopolymers, such as collagen [17] and blends of bio-based polymers, including methylcellulose–chitosan composites [18]. Because of the structure containing three hydroxyl groups and a carboxylic moiety, gallic acid was also exploited in order to receive a bio-based epoxy thermoset [19]. Moreover, the therapeutical properties of gallic acid, including bone regeneration improving or inhibition of enamel demineralization, allow to dedicate it to dental applications [20] as well as to non-toxic nanoparticles preparation for biomedical use [21]. Therefore, such a compound can be added to materials to modify them in a safe way.

Similar properties were detected in the case of tannin acid, which consists of a glucose unit in the center of the structure with ten gallic acid molecules attached to it (Figure 2) [22]. Unlike gallic acid, tannic acid does not contain carboxyl groups. This polyphenolic compound of high molecular weight can be extracted with high efficiency from herbaceous as well as woody plants. Tannic acid is famous for its antiviral and antibacterial properties, but its cross-linking effectiveness is also attractive from the polymer technology point of view. So far, this phenolic acid has been studied in combination with natural polymers, including collagen, chitosan, starch, agarose, hyaluronic acid as well as silk. Moreover, tannic acid was utilized as a crosslinker for catalyst-free silicone elastomers and to cure epoxidized soybean oil to obtain fully bio-based epoxy resin [23]. It was also proved that this acid is able to cure UV-treated epoxidized natural rubber to receive environmentally friendly films dedicated to coatings [24]. Tannic acid is definitely an interesting modifier, which can be utilized to obtain biomaterials demonstrating antiviral and antibacterial properties.

## 2. Materials and Methods

The studied materials were prepared by mixing 100 phr (parts per hundred parts of rubber) of ENR rubber containing 50 mol% epoxidation purchased from Muang Mai Guthrie Company Limited (Phuket, Thailand) named Dynathai Epoxyprene 50 (ENR-50) with 30 phr of hydrophilic fumed silica (Aerosil 380) characterized by a specific surface area of 380 m^2^/g obtained from Evonik Operations GmbH (Essen, Germany). The selected natural phenolic compounds, including quercetin, gallic acid, and tannic acid as modifiers were added in a proportion of 3 phr. Quercetin hydrate (≥95% of purity) was purchased from Sigma-Aldrich (Munich, Germany), tannic acid (≥93% of purity) produced by Pol-Aura (Olsztyn, Poland) and gallic acid 1-hydrate (≥99% of purity) was obtained from Pol-Aura (Olsztyn, Poland).

The mixing process was conducted using a laboratory mixing mill (David Bridge & co, Rochdale, Great Britain) with a friction of 1–1.2 at ambient temperature for about 10 min. The prepared unvulcanized mixtures were first tested using an Alpha MDR 2000 oscillating disc rheometer (Alpha Technologies, Hudson, OH, USA) to assess the curing properties. The measurement was performed at the temperature of 160 °C for 60 min and the curing curve was obtained. According to the results, we decided to prepare our vulcanizates within 20 min at a temperature of 160 °C using the pressure of 14 MPa of an electrically heated hydraulic laboratory press (Skamet 54436, SKAMET, Skarzysko-Kamienna, Poland) to achieve semi-vulcanized materials with satisfactory mechanical properties prone to reprocess. Therefore, the torque at 20th min (M_20_) and minima torque (M_min_) were analyzed and used to calculate a torque increase at the 20th min (dM_20_ = M_20_ − M_min_).

The vulcanizates were rectangular as they had been formed in special steel vulcanization molds situated between the press shelves, 120 mm long, 80 mm wide and 1 mm thick. The polytetrafluoroethylene (PTFE) spacers produced by Holtex^®^ (Rzgow, Poland) prevented adhesion. The prepared composites were analyzed in terms of different mechanical properties, color, and chemical structures.

Firstly, the prepared samples were formed into dumbbell-shaped specimens (type 2 according to PN-ISO 37:1998 standard) by cutting with a special stamp. Such specimens were 1 mm thick, 75 mm long, and 12.5 mm wide at ends and were useful for performing static as well as dynamic mechanical analysis. The mechanical analysis was carried out utilizing the universal mechanical testing machine Zwick 1435 (Zwick Roell GmbH & Co. KG, Ulm, Germany). Each of the 5 specimens was stretching at a crosshead speed of 500 mm/min according to PN-ISO 37:1998 standard. The most important information, including tensile strength (TS) and elongation at break (E_b_), was analyzed.

Two of the rectangular ENR-based samples were cut into small pieces and reprocessed at the same conditions as the pristine samples (1st part) as well as using a higher temperature (180 °C) (2nd part) and then also cut into dumbbell-shaped specimens and subsequently retested using static mechanical analysis. The effectiveness of reprocessing was calculated using tensile strength results and the formula below:(1)$$R_{\mathrm{TS}}\left(\%\right)=\frac{{\mathrm{TS}}_{\mathrm{after}\;\mathrm{reprocessing}}}{{\mathrm{TS}}_{\mathrm{pristine}}}\times 100\%$$
where:R_TS_ (%)—effectiveness of reprocessing calculated using tensile strength results [%],TS_after reprocessing_—tensile strength of material after reprocessing [MPa],TS_pristine_—tensile strength of pristine material [MPa].

The effectiveness of reprocessing was also calculated using elongation at break (Eb) results and the formula below:(2)$$R_{\mathrm{Eb}}\left(\%\right)=\frac{{\mathrm{Eb}}_{\mathrm{after}\;\mathrm{reprocessing}}}{{\mathrm{Eb}}_{\mathrm{pristine}}}\times 100\%$$
where:R_Eb_ (%)—effectiveness of reprocessing [%],Eb_after reprocessing_—elongation at break of material after reprocessing [%],Eb_pristine_—elongation at break of pristine material [%].

Dynamic mechanical analysis was performed in tension mode applying a DMA/SDTA861e analyzer (Mettler Toledo, Greifensee, Switzerland). The investigation was conducted in a temperature range of −80–85 °C with a heating rate of 3 °C/min, a frequency of approximately 5 Hz, and a strain amplitude of 4 μm. The most important parameters, including storage (E′) and loss (E″) moduli, damping factor (tanδ), were determined during the measurement.

The chemical structure of elastomeric bio-composites was investigated with a Thermo Scientific Nicolet 6700 Fourier transform infrared spectroscopy (FT-IR) device equipped with a diamond Smart Orbit ATR sampling accessory (Thermo Fischer Scientific Instruments, Waltham, MA, USA). The analysis was carried out in absorption mode, where the spectra were obtained in the range of 4000–400 cm^−1^ with the use of 64 scans and a resolution of 4 cm^−1^.

The impact of the added natural phenolic compounds on the color of ENR/silica hybrids was described using the CIE-Lab system, where the L-axis corresponds to lightness, the a-axis reveals red–green tone expression, and the b-axis is responsible for yellow–blue colors determination. The UV-VIS CM-36001 spectrophotometer (Konica Minolta Sensing, Inc., Osaka, Japan) in accordance with the PN-EN ISO 105-J01 standard was utilized to detect and convert the signal reflected from the surface into proper parameters. Such parameters, including a (red-green tones), b (yellow-blue tones), and L (lightness) parameters, were used for calculating the color difference (dE) between ENR/silica composites containing natural phenolic compounds and the pure ENR/silica hybrid based on Equation (3).
(3)$$E=\sqrt{\Delta a^{2}+\Delta b^{2}+\Delta L^{2}}$$
where:Δa—difference of a parameter between samples with natural additive and pure ENR/silica hybrid;Δb—difference of b parameter between samples with natural additive and pure ENR/silica hybridΔL—difference of L parameter between samples with natural additive and pure ENR/silica hybrid.

Moreover, the whiteness index (W_i_), chroma (C_ab_), and hue angle (h_ab_) parameters of ENR/silica composites with quercetin, tannin acid, and gallic acid as well as ENR with only silica were calculated based on Equations (4)–(6).
(4)$$W_{i}=100-\sqrt{a^{2}+b^{2}+{\left(100-L\right)}^{2}}$$
(5)$$C_{\mathrm{ab}}=\sqrt{a^{2}+b^{2}}$$
(6)$$h_{\mathrm{ab}}\{\begin{array}{c} \mathrm{arctg}\left(\frac{b}{a}\right),\;\mathrm{when}\;a>0\;\cap^{\;}b>0\\ 180{}^{\circ}+\mathrm{arctg}\left(\frac{b}{a}\right),\;\mathrm{when}\;\left(a<0\cap^{\;}b>0\right)\cup^{\;}(a<0\cap^{\;}b<0)\\ 360{}^{\circ}+\mathrm{arctg}\left(\frac{b}{a}\right),\;\mathrm{when}\;a>0\cap^{\;}b<0 \end{array}$$

Additionally, in order to analyze the appearance of the prepared composites, the samples’ surface was studied using a Leica MZ6 stereoscopic microscope (Heerbrugg, Switzerland) equipped with MultiScan 8.0 image analysis software (CSS, Warsaw, Poland) at a magnification of 50×.

## 3. Results and Discussion

### 3.1. Curing Characteristics

The crosslinking effect can be easily observed in a curing curve (Figure 3a), where an increase in torque as a function of curing time takes place. Such an effect is visible in the case of ENR-based composites containing only silica, which was proved by Xu et al. [9]. According to Figure 3a,b, the referential sample containing only 30 phr of silica revealed a slight increase in torque, which amounted to 0.5 dNm at 20th min of vulcanizing. Interestingly, the addition of 3 phr of natural phenolic compounds, including quercetin and tannic acid, caused a more visible increase in dM_20_ to 1.2 dNm and 1.7 dNm, respectively. The most significant changes occurred after adding 3 phr of gallic acid, where the increase in torque at the 20th min of curing amounted to 3.6 dNm. Such relevant changes resulted from the structure of gallic acid containing not only hydroxyl groups but also a carboxylic moiety, which is able to react easily with the oxirane ring belonging to the epoxidized natural rubber chain as well as with hydroxyl groups situated in silica aggregates. The minimal torque (M_min_) indicates the ease of processing. Materials with the lowest M_min_, including ENR/SIL/QUE and ENR/SIL/TA, are characterized by a more suitable mixing performance of compound than the referential sample. On the other hand, the addition of 3 phr of gallic acid caused an increase in minimal torque from 1.1 dNm to 1.4 dNm, therefore, slightly deteriorating the processing of the ENR/silica material. Moreover, the curing curves after adding natural phenolic compounds to the ENR/SIL material are typical for the marching type. It means that the longer the curing time, the higher the increase in torque (dM). It was decided to prepare all types of ENR/SIL composites by pressing at 20 min, as this is sufficient time for obtaining the best mechanical properties of the referential sample. More precisely, we wanted to observe the impact of natural phenolic compounds on the same properties of identically prepared ENR/silica composites.

### 3.2. Mechanical and Reprocessing Properties

The results obtained from the static mechanical analysis are presented in Table 1 and Table 2 and Figure 4a. According to the results, the impact of the selected natural phenolic compound on tensile strength (TS), elongation at break (Eb), and reprocessing effectiveness (R_TS_, R_Eb_) was significantly noticeable after the addition of 3 phr of gallic acid. This phenolic acid caused the most relevant changes in tensile strength, which increased from 2.65 ± 0.08 MPa to about 9.5 ± 0.3 MPa, slightly improving also the elongation at break from 301 ± 7% to 357 ± 14%. A triple hydroxyl group and carboxyl moiety contributed to such an enhancement. Mixing quercetin as well as tannic acid with the ENR/silica hybrid also resulted in an improvement in TS and Eb. Quercetin, thanks to the phenolic groups’ presence, was able to improve the tensile strength of ENR/silica hybrid from 2.65 ± 0.08 MPa to 5.8 ± 0.2 MPa and remarkably boosted elongation at break from 301 ± 7% to 500 ± 20%. On the other hand, tannic acid also contributed to the enhancement of the ENR/silica hybrid, and TS amounted to 5.70 ± 0.18 MPa, but only a slight improvement in Eb was noticeable (Eb = 368 ± 4%). Semi-crosslinked composites seemed to be reprocessable; nevertheless, only the referential sample achieved satisfactory effectiveness after reprocessing at 160 °C. Interestingly, recycling composites at a higher temperature (T = 180 °C) caused satisfying tensile strength of the ENR/TA composite after this process, where the effectiveness of reprocessing amounted to 106%. Moreover, ENR/SIL/QUE also regained TS after the recycling process but with a slightly lower effectiveness of 95%, which is also a satisfactory result. Unfortunately, ENR/SIL/GA composites are materials with probably the highest and strongest intermolecular crosslinks and interactions; therefore, their recycling is not possible with high effectiveness. It must also be underlined that in spite of the high effectiveness of reprocessing, calculated based on the TS results, these composites are able to regain only a half of their elongation at break.

The results of the dynamic mechanical analysis are presented in Figure 4b–d.

Dynamic mechanical thermal analysis was useful for observing the changes in three parameters as a function of temperature: storage modulus (E′) related to stored energy and representing the elastic portion, loss modulus (E″) referred to the energy dissipated as heat, which corresponded to the viscous portion. Additionally, the damping factor (tanδ) was also observed. The damping factor is defined as tanδ = E″/E′ and indicates the energy dissipation of a material along with enabling the detection of the glass transition temperature (Tg). In this research, the glass transition temperature was detected as a temperature at a maximum damping factor (tanδ). All curves belonging to the tested ENR/SIL composites with natural phenolic compounds are presented in Figure 4b–d as well as in Table 3. As shown in Figure 4b, in the range of −60 °C to 20 °C, considerable changes between tested composites are revealed. It was noticed that the storage modulus (E′) of the pure ENR/SIL composite and composites with natural phenolic compounds was found to decrease as a function of temperature because of the softening of the polymer chains with the temperature. The raised storage modulus for the ENR/silica composites containing natural phenolic compounds occurred particularly in the case of ENR/SIL/GA due to the reinforcement and the mobility restriction as a result of a crosslinking effect. Moreover, aromatic rings, which belong to natural phenolic compounds are rigid and might contribute to the whole material stiffening. Natural phenolic compounds also caused the material’s glass transition temperature increase from −9.7 °C to −7.7 °C in the case of tannic acid, to −5.4 °C after adding quercetin and to −1.9 °C due to gallic acid. Such a phenomenon also confirmed the stiffening effect of the ENR/SIL material.

### 3.3. Chemical Structure Analysis

According to the spectrum depicted in Figure 5, the ENR/silica composites are characterized by the intensive peak visible at 1069 cm^−1^ of wavenumber, which probably belongs to three types of moieties present in the structure, i.e., Si-O-Si asymmetrical stretching (1099 cm^−1^) [9], aromatic C-H in-plane bending (1096 cm^−1^) [25] as well as C-O stretching vibrations [25]. The intensity of such a band depends on natural phenolic compound addition. The highest peak belongs to the ENR/silica composite with gallic acid, and the lowest is visible in the ENR/silica composite with gallic acid. In comparison to pure unvulcanized ENR, ENR/silica elastomers have a band at 798 cm^−1^ of wavenumber, which likely corresponds to aromatic out-of-plane C-H bending vibrations (793 cm^−1^) [26] and can belong to symmetrical stretching vibrations of Si-O (795 cm^−1^) [9]. The most important differences in chemical structure between the tested composites with different natural phenolic additives were noticeable in the spectrum between 1500 and 1750 cm^−1^ of wavenumber. In the spectrum of the ENR/SIL/QUE composite, the characteristic band occurring at 1654 cm^−1^ is related to stretching vibrations of C=C [9] and can correspond with a C=O stretch in enol form [27]. On the other hand, the spectrum of ENR/SIL/TA reveals an intensive band at 1590 cm^−1^ and 1599 cm^−1^, which is characteristic for stretching vibrations of aromatic C=C + C-O groups belonging to tannins [25]. In ENR/SIL/GA acid, the characteristic band is visible at 1708 cm^−1^ [28]. Adding only silica to pure ENR-50 and vulcanizing at 160 °C caused a reduction in peaks at 1577 cm^−1^ and 1540 cm^−1^, which correspond with carboxyl symmetric stretching vibrations [29]. It is noticed that most of the visible bands came from ENR or silica and only range from 1500–1750 cm^−1^ enabling discernment of the difference between hybrids containing individually applied natural phenolic compounds.

### 3.4. Color Assessment

Natural phenolic compounds can also contribute to a material’s color change, which is observable in Figure 6 and Figure 7. It is assumed that a color difference (dE) above 5 indicates the detection of two different colors. According to Figure 6a, all natural compounds caused a color modification but the most intensive color change of about 23 was noticeable in the case of quercetin addition. The different colors of prepared samples are also visible in Figure 6d and Figure 7a–d. Based on Figure 6b, the ENR/SIL/QUE composite has the lowest whiteness index, which amounted to below 40. The lower whiteness index indicates that the hybrid became darker after adding quercetin. On the other hand, the addition of tannic acid and gallic acid resulted in a similar color change of about 17. Such composites are characterized not only by a lower whiteness index but also by a higher chroma than the pure ENR/SIL composite. The higher chroma indicates that the color of these hybrids is more intense. Moreover, slight differences are also visible in the hue angle (h_ab_) results (Figure 6c). The samples with a higher angle of 80°–100° represent more yellow tones but lower hue angles below 80° received reddish tones because of the compound. This means that natural phenolic compounds can act also as a gentle natural colorant dedicated to ENR/silica hybrids.

Based on Figure 7, the silica aggregates are visible in the case of all composites, but natural compounds caused less transparency in these materials. According to the results, the selected natural phenolic compounds can be additionally used as colorants.

## 4. Conclusions

ENR/silica composites are pro-ecologic and simply prepared materials. The addition of natural phenolic compounds, which are safe ingredients of plant origin, yields environmentally friendly biomaterials with enhanced mechanical properties in comparison to pure ENR/SIL. The incorporation of 3 phr of quercetin caused an increase in tensile strength from 2.65 ± 0.08 MPa to 5.8 MPa and elongation at break from 301 ± 7% to 500 ± 20%. Adding the same amount of tannic acid also provided higher values of TS (5.70 ± 0.18 MPa) but without significant changes in Eb. Gallic acid dramatically changed the tensile strength to 9.5 ± 0.3 MPa. The selected natural phenolic compounds in combination with silica act as crosslinking agents and cause an increase in torque. Moreover, these substances are able to elevate the glass transition temperature to higher values and slightly change the color of ENR/SIL composites. Such composites with natural ingredients have a chance to become desirable biomaterials in the biomedical sector.

## Figures and Tables

**Figure 1 molecules-27-02214-f001:**
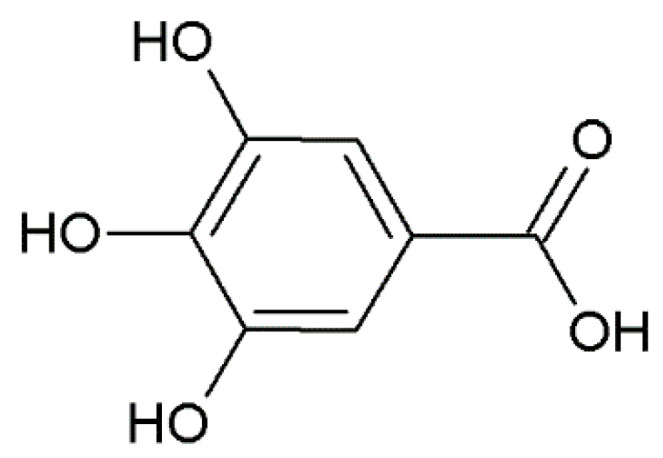
Chemical structure of gallic acid.

**Figure 2 molecules-27-02214-f002:**
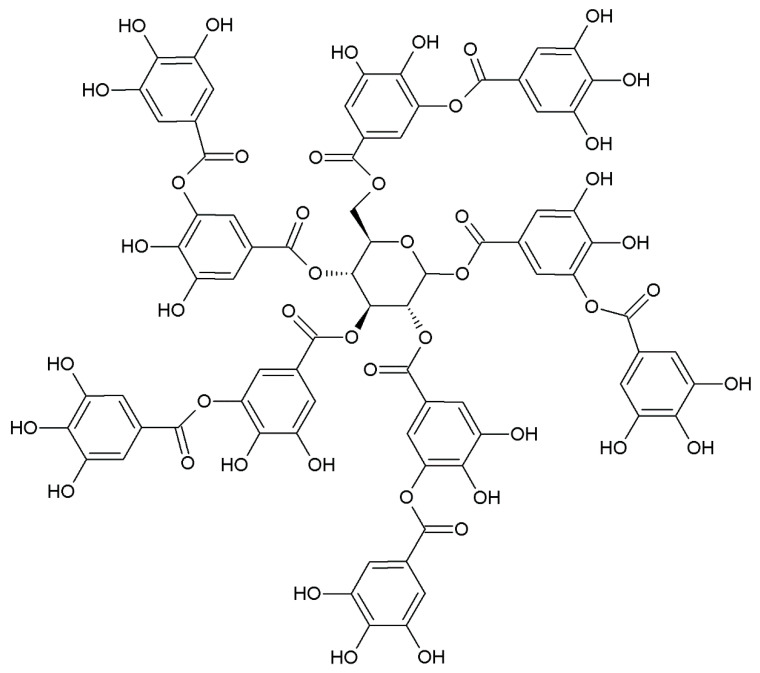
Chemical structure of tannic acid.

**Figure 3 molecules-27-02214-f003:**
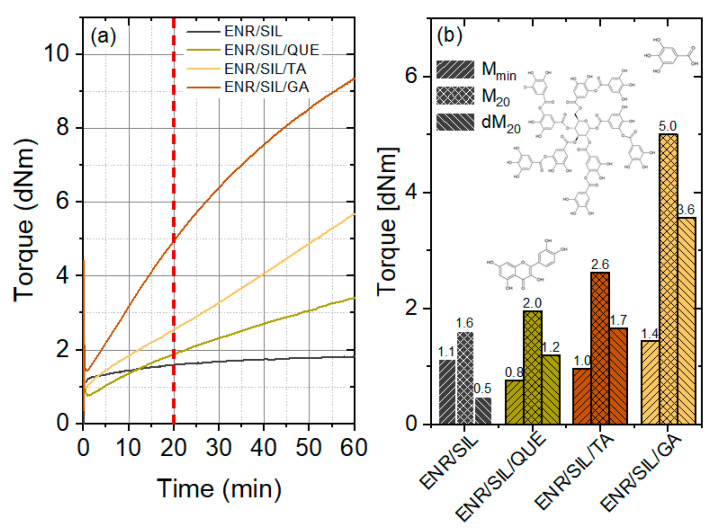
Analysis of curing effectiveness of ENR/silica materials containing natural phenolic compounds as a function of time (**a**) and values of the minimal torque (Mmin), torque at 20th min of vulcanizing (M_20_), and torque increase at 20th min of vulcanizing (dM_20_) of the same materials (**b**).

**Figure 4 molecules-27-02214-f004:**
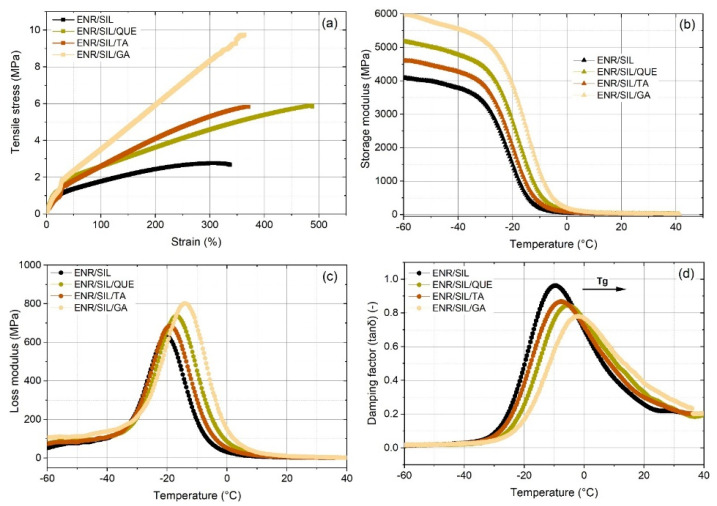
The selected strain-stress curves of tested ENR/silica composites with natural phenolic compounds obtained from static mechanical analysis (**a**), storage modulus results (**b**), loss modulus results (**c**), and damping factor (**d**) of the same composites obtained from dynamic mechanical analysis (DMA).

**Figure 5 molecules-27-02214-f005:**
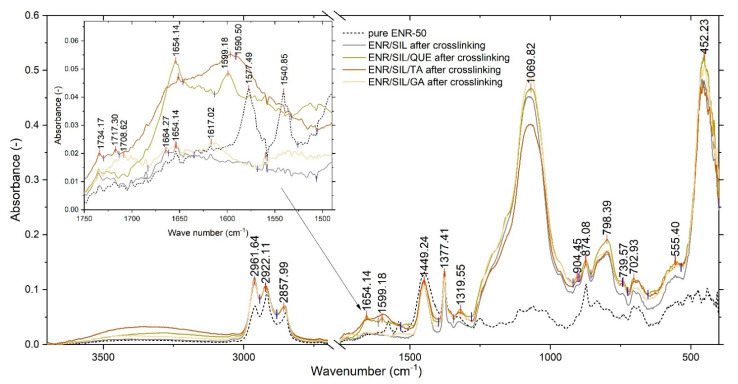
Comparison spectrum of ENR/silica hybrid containing natural phenolic compounds with spectra of pure ENR/silica hybrid.

**Figure 6 molecules-27-02214-f006:**
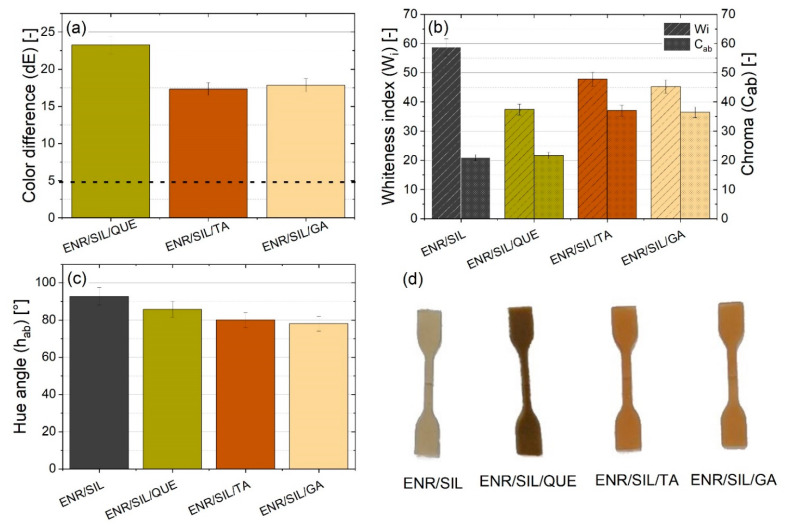
The color difference between ENR/SIL composite and ENR/SIL material with natural phenolic compounds addition (**a**), whiteness index and chroma (**b**), hue angle (**c**), and photos of ENR/silica-based samples (**d**).

**Figure 7 molecules-27-02214-f007:**
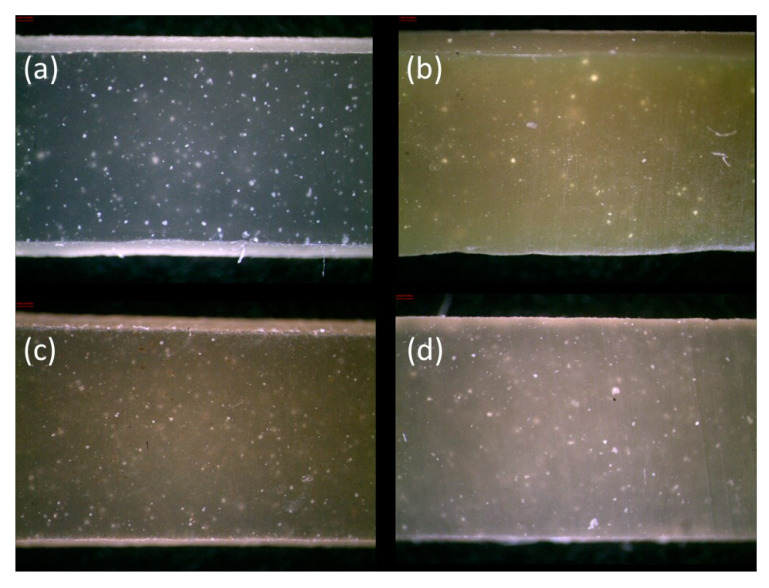
The microscopic photos revealing appearance of pure ENR/SIL composite and ENR/SIL material with natural phenolic compounds addition: pure ENR/silica sample (**a**), ENR/silica hybrid with quercetin (**b**), ENR/silica hybrid with tannic acid (**c**), ENR/silica hybrid with gallic acid (**d**).

**Table 1 molecules-27-02214-t001:** Tensile strength (TS) results of ENR/silica composites with natural phenolic compounds before and after reprocessing at 160 °C and 180 °C with effectiveness of reprocessing calculated using tensile strength results (R_TS_ (%)).

Sample	TS of Pristine Sample [MPa]	TS of Reprocessed Sample at 160 °C [MPa]	R_TS_ of Reprocessed Sample at 160 °C [%]	TS of Reprocessed Sample at 180 °C [MPa]	R_TS_ of Reprocessed Sample at 180 °C [%]
ENR/SIL	2.65 ± 0.08	2.8 ± 0.1	105	1.9 ± 0.4	71
ENR/SIL/QUE	5.8 ± 0.2	2.5 ± 0.9	42	5.5 ± 0.6	95
ENR/SIL/TA	5.70 ± 0.18	3.7 ± 0.8	65	6.0 ± 0.7	106
ENR/SIL/GA	9.5 ± 0.3	1.21 ± 1.24	13	1.5 ± 0.5	16

**Table 2 molecules-27-02214-t002:** Elongation at break (Eb) results of ENR/silica composites with natural phenolic com-pounds before and after reprocessing at 160 °C and 180 °C with effectiveness of reprocessing calculated using elongation at break results (R_Eb_ (%)).

Sample	E_b_ of Pristine Sample [MPa]	E_b_ of Reprocessed Sample at 160 °C [MPa]	R_Eb_ of Reprocessed Sample at 160 °C [%]	E_b_ of Reprocessed Sample at 180 °C [MPa]	R_Eb_ of Reprocessed Sample at 180 °C [%]
ENR/SIL	301 ± 7	270 ± 20	90	150 ± 70	49
ENR/SIL/QUE	500 ± 20	120 ± 50	24	230 ± 30	46
ENR/SIL/TA	368 ± 4	140 ± 40	39	190 ± 20	52
ENR/SIL/GA	357 ± 14	33 ± 19	9	40 ± 11	11

**Table 3 molecules-27-02214-t003:** Additional results showing the changes of properties in temperature for ENR/silica composites containing selected natural phenolic compounds and the temperature values corresponding to the maxima detected for loss modulus and damping factor.

Sample	Storage Modulus Changes in Temperature						T_max E″_ [°C]	T_max tanδ_ (T_g_) [°C]
	E′_−60_ [MPa]	E′_−40_ [MPa]	E′_−20_ [MPa]	E′_0_ [MPa]	E′_20_ [MPa]	E′_40_ [MPa]		
ENR/SIL	4083.0	3779.5	1473.5	42.4	13.7	8.8	−20.3	−9.7
ENR/SIL/QUE	5176.3	4777.5	2764.1	107.5	22.4	12.2	−16.8	−5.4
ENR/SIL/TA	4593.4	4267.3	1990.4	72.5	17.3	9.5	−19.1	−7.7
ENR/SIL/GA	5979.7	5543.2	3844.3	185.8	26.3	9.5	−14.0	−1.9

T_max E″_—temperature at which the maximum of loss modulus is observed, T_max tanδ_—temperature at which the maximum of damping factor is detected.

## Data Availability

Data sharing is not applicable for this article.

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
