# Peer review of "Natural Phenolic Compounds as Modifiers for Epoxidized Natural Rubber/Silica Hybrids"

_molecules, 2022, doi:10.3390/molecules27072214_

Round 1

Reviewer 1 Report

This paper investigates the effect of adding natural phenolic compounds to epoxidized natural rubber/silica composite systems in terms of mechanical properties and color changes. There seem to be a number of typographical errors and insufficient explanations. Please reconsider the following points.

  • Experimental section: Equations 4-6 are missing.
  • Figure 3: “ML” in the legend should be “Mmin”.
  • Regarding the reprocessing experiment, the samples' R-values mixed with phenol are all higher at 180 degrees C than at 160 degrees C. Is it because of the increase of binding sites? If IR data or other data can show this situation, such a discussion should be added.
  • Line 244: Figure 2 (b) -> Figure 4 (b)
  • The authors take the peak temperature of tan δ measured DMA at 5 Hz as Tg without any explanation, but it should be explained that this is an assumption made by the authors.
  • IR spectra: It is difficult to understand what was found from the comparison of the IR spectra. Please concisely summarize the authors’ points.
  • Line291: Figure 4 (a) -> Figure 6 (a)
  • Line 293: FIgure 5 (b) -> Figure 6 (b)
  • Line 297: Figure 5 -> Figure 7
  • Color assessment section: I am not sure what conclusions can be drawn from Figures 6 (a),(b),(c) since there is no explanation of the indices representing the colors. Perhaps the general reader will not get what the authors want to say. Please explain the advantages of using these indexes. It would also be necessary to explain whether the coloring phenomenon is caused by compounding or whether it is just natural phenolic components that are originally colored.

Author Response

Institute of Polymer and Dye Technology

Technical University of Lodz

90-924 Lodz, ul Stefanowskiego 12/16, Poland

Tel.: +48 42 631 32 23, Fax: +48 42 636 25 43

March 20, 2022

Molecules

Dear Professor,

We are resubmitting our revised paper entitled Natural Phenolic Compounds as Modifiers for Epoxidized Natural Rubber/Silica Hybrids by Anna Masek and Olga Olejnik with a request to reconsider it for publication in Molecules. We have carefully considered the Editor and Reviewers' comments. The manuscript was revised exactly according to these comments. The list of responses to the reviewers’ comments and corrections made in the manuscript is attached.

The manuscript has not been previously published, is not currently submitted for review to any other journal, and will not be submitted elsewhere before a decision is made by this journal.

For correspondence please use the following information:

corresponding author: Anna Masek

Institute of Polymer and Dye Technology

Technical University of Lodz

90-924 Lodz, ul Stefanowskiego 12/16, Poland

Tel.: +48 42 631 32 93

Fax: +48 42 636 25 43

e-mail: anna.masek@p.lodz.pl

Yours sincerely,

Ph. D., D.Sc. Anna Masek

Reviewer #1:

This paper investigates the effect of adding natural phenolic compounds to epoxidized natural rubber/silica composite systems in terms of mechanical properties and color changes. There seem to be a number of typographical errors and insufficient explanations. Please reconsider the following points.

  • Experimental section: Equations 4-6 are missing..

Answer 1 for Reviewer #1:

We thank the Reviewer for paying attention to missing equations. The equations must have disappeared during the final edition. We have already corrected this mistake as follows:  

Such parameters, including: a (red-green tones), b (yellow-blue tones) and L (lightness) parameters were used for calculating the color difference (dE) between ENR/silica composites containing natural phenolic compounds and pure ENR/silica hybrid based on equation 3.

                                                       (3)

where:

    Moreover, the whiteness index (Wi), chroma (Cab) and hue angle (hab) parameters of ENR/silica composites with quercetin, tannin acid and gallic acid as well as ENR with only silica were calculated based on equations 4-6.

                     (4)

                                                 (5)

  (6)

Reviewer #1:

  • Figure 3: “ML” in the legend should be “Mmin”.

Answer 2 for Reviewer #1:

We are grateful that Reviewer noticed our mistake. Figure 3 has been improved as follows:

Reviewer #1:

Reviewer #1:

  • Regarding the reprocessing experiment, the samples' R-values mixed with phenol are all higher at 180 degrees C than at 160 degrees C. Is it because of the increase of binding sites? If IR data or other data can show this situation, such a discussion should be added.

Answer 3 for Reviewer #1:

We appreciate the Reviewer’s suggestions. Nevertheless, the reprocessing experiment was only additional and interesting option and we are mainly focused on pristine materials. Moreover, FT-IR spectra reveal bands which mostly come from ENR or silica and binding sites involved in crosslinking-effect, especially the quantity of them, is hard to observe. In our FT-IR spectra analysis we are mostly focused on qualitative analysis.

Reviewer #1:

  • Line 244: Figure 2 (b) -> Figure 4 (b)

Answer 4 for Reviewer #1:

We are thankful for the Reviewer’s comment. The manuscript had been modified a fiew times, therefore the mistake occurred. We have already corrected the mistake.

Reviewer #1:

  • The authors take the peak temperature of tan δ measured DMA at 5 Hz as Tg without any explanation, but it should be explained that this is an assumption made by the authors.

Answer 5 for Reviewer #1:

We appreciate Reviewer’s suggestions. We have already added information as follows: “In this research glass transition temperature was detected as a temperature at a maximum of damping factor (tanδ).”

Reviewer #1:

  • Line291: Figure 4 (a) -> Figure 6 (a)
  • Line 293: FIgure 5 (b) -> Figure 6 (b)
  • Line 297: Figure 5 -> Figure 7

Answer 6 for Reviewer #1:

We are grateful for the Reviewer’s comment. As we mentioned, the manuscript had been modified a few times, therefore the mistakes occurred. We have already corrected the mistakes.

Reviewer #1:

  • IR spectra: It is difficult to understand what was found from the comparison of the IR spectra. Please concisely summarize the authors’ points.

Answer 7 for Reviewer #1:

We appreciate Reviewer’s suggestions. FT-IR spectra analysis was relevant to characterize structure of the prepared materials. To summarize our characterization: it is noticed, that most of visible bands came from ENR or silica and only range from 1500-1750 cm-1 enable telling the difference between hybrids containing individually applied natural phenolic compound.

Reviewer #1:

  • Color assessment section: I am not sure what conclusions can be drawn from Figures 6 (a),(b),(c) since there is no explanation of the indices representing the colors. Perhaps the general reader will not get what the authors want to say. Please explain the advantages of using these indexes. It would also be necessary to explain whether the coloring phenomenon is caused by compounding or whether it is just natural phenolic components that are originally colored.

Answer 8 for Reviewer #1:

We thank the Reviewer for this suggestion. We change our color description as follows:

Natural phenolic compounds can also contribute to material’s color changing, which is observable in Figure 6 and Figure 7. It is assumed that color difference (dE) above 5 indicates a detection of two different colors. According to Figure 6 (a) all natural compounds caused a color modification but the most intensive color changing of about 23 was noticeable in the case of quercetin addition. Based on Figure 6 (b) ENR/SIL/QUE composite has the lowest whiteness index, which amounted to below 40. The lower whiteness index indicates, that the hybrid become darker after adding quercetin. On the other hand, an addition of tannic acid and gallic acid resulted in similar color changing of about 17. Such composites are characterized not only by lower whiteness index but also by higher chroma than pure ENR/SIL composite. The higher chroma indicates, that the color of these hybrids is more intensive. Moreover, the slight differences are also visible in hue angle (hab) results (Figure 6 (c)). The samples with higher angle of 80°-100° represent more yellow tones but lower hue angle below 80° received reddish tones because of the compound. It means, that natural phenolic compounds can act also as a gentle natural colorant dedicated to ENR/silica hybrids.

Reviewer 2 Report

In this manuscript, Anna Masek et al. proposed natural phenolic compounds, including quercetin, tannic acid and gallic acid to modify epoxidized natural rubber/silica hybrids for obtaining bio-elastomers with improved mechanical properties. The modifiers of Epoxidized Natural Rubber/Silica Hybrids by Natural Phenolic Compounds have been widely reported and studied [Polymer Testing, 2016, 54: 176-185; Polymer Testing, 2018, 67: 92-98; Polymers, 2019, 11(11): 1763; Polymer Degradation and Stability, 2021, 185: 109482.]. However, after careful reading of the manuscript, I could not recommend publication of this manuscript in Molecules since the is no new insights or novel findings to the level of this journal. The reason for my decision is provided below:

  1. This work does not provide new fundamental insights.
  2. Where are quations 4-6 in 166 lines?
  3. There are some typos in the manuscript need to be revised. Also, the authors should carefully check English grammar.

Author Response

Institute of Polymer and Dye Technology

Technical University of Lodz

90-924 Lodz, ul Stefanowskiego 12/16, Poland

Tel.: +48 42 631 32 23, Fax: +48 42 636 25 43

March 20, 2022

Molecules

Dear Professor,

We are resubmitting our revised paper entitled Natural Phenolic Compounds as Modifiers for Epoxidized Natural Rubber/Silica Hybrids by Anna Masek and Olga Olejnik with a request to reconsider it for publication in Molecules. We have carefully considered the Editor and Reviewers' comments. The manuscript was revised exactly according to these comments. The list of responses to the reviewers’ comments and corrections made in the manuscript is attached.

The manuscript has not been previously published, is not currently submitted for review to any other journal, and will not be submitted elsewhere before a decision is made by this journal.

For correspondence please use the following information:

corresponding author: Anna Masek

Institute of Polymer and Dye Technology

Technical University of Lodz

90-924 Lodz, ul Stefanowskiego 12/16, Poland

Tel.: +48 42 631 32 93

Fax: +48 42 636 25 43

e-mail: anna.masek@p.lodz.pl

Yours sincerely,

Ph. D., D.Sc. Anna Masek

Reviewer #1:

Comments and Suggestions for Authors

In this manuscript, Anna Masek et al. proposed natural phenolic compounds, including quercetin, tannic acid and gallic acid to modify epoxidized natural rubber/silica hybrids for obtaining bio-elastomers with improved mechanical properties. The modifiers of Epoxidized Natural Rubber/Silica Hybrids by Natural Phenolic Compounds have been widely reported and studied [Polymer Testing, 2016, 54: 176-185; Polymer Testing, 2018, 67: 92-98; Polymers, 2019, 11(11): 1763; Polymer Degradation and Stability, 2021, 185: 109482.]. However, after careful reading of the manuscript, I could not recommend publication of this manuscript in Molecules since the is no new insights or novel findings to the level of this journal. The reason for my decision is provided below:

  1. This work does not provide new fundamental insights.

Answer 1 for Reviewer #1:

We thank Reviewer for these suggestions. We have changed abstract to present fundamental insights. The most important is cooperation of silica with natural phenolic compounds which provide improved crosslinking effect in comparison to pure silica. It is important to eliminate toxic crosslinkers, including dicumyl peroxide to obtain totally safe biomaterials. It is important, that only 3 phr of natural phenolic compound can provide better crosslinking effect of ENR/silica hybrids without any toxic compounds.

Reviewer #1:

  1. Where are quations 4-6 in 166 lines?

We thank the Reviewer for paying attention to missing equations. The equations must have disappeared during the final edition. We have already corrected this mistake as follows:

Such parameters, including: a (red-green tones), b (yellow-blue tones) and L (lightness) parameters were used for calculating the color difference (dE) between ENR/silica composites containing natural phenolic compounds and pure ENR/silica hybrid based on equation 3.

                                                      (3)

where:

    Moreover, the whiteness index (Wi), chroma (Cab) and hue angle (hab) parameters of ENR/silica composites with quercetin, tannin acid and gallic acid as well as ENR with only silica were calculated based on equations 4-6.

                     (4)

                                                 (5)

(6)

Reviewer #1:

  1. There are some typos in the manuscript need to be revised. Also, the authors should carefully check English grammar.

Answer 1 for Reviewer #1:

We thank the Reviewer for paying attention to English grammar. We have improved our manuscript and corrected grammatical errors.

@font-face {font-family:"Cambria Math"; panose-1:2 4 5 3 5 4 6 3 2 4; mso-font-charset:0; mso-generic-font-family:roman; mso-font-pitch:variable; mso-font-signature:-536870145 1107305727 0 0 415 0;}@font-face {font-family:Calibri; panose-1:2 15 5 2 2 2 4 3 2 4; mso-font-charset:238; mso-generic-font-family:swiss; mso-font-pitch:variable; mso-font-signature:-469750017 -1073732485 9 0 511 0;}@font-face {font-family:"Palatino Linotype"; panose-1:2 4 5 2 5 5 5 3 3 4; mso-font-charset:0; mso-generic-font-family:roman; mso-font-pitch:variable; mso-font-signature:-536870265 1073741843 0 0 415 0;}p.MsoNormal, li.MsoNormal, div.MsoNormal {mso-style-unhide:no; mso-style-qformat:yes; mso-style-parent:""; margin-top:0cm; margin-right:0cm; margin-bottom:10.0pt; margin-left:0cm; line-height:115%; mso-pagination:widow-orphan; mso-hyphenate:none; font-size:11.0pt; font-family:"Calibri",sans-serif; mso-fareast-font-family:Calibri; mso-bidi-font-family:"Times New Roman"; mso-font-kerning:1.0pt; mso-fareast-language:AR-SA;}p.MDPI31text, li.MDPI31text, div.MDPI31text {mso-style-name:"MDPI_3\.1_text"; mso-style-unhide:no; mso-style-qformat:yes; mso-style-parent:""; margin-top:0cm; margin-right:0cm; margin-bottom:0cm; margin-left:130.4pt; text-align:justify; text-indent:21.25pt; line-height:95%; mso-pagination:widow-orphan; layout-grid-mode:char; mso-layout-grid-align:none; font-size:10.0pt; mso-bidi-font-size:11.0pt; font-family:"Palatino Linotype",serif; mso-fareast-font-family:"Times New Roman"; mso-bidi-font-family:"Times New Roman"; color:black; mso-ansi-language:EN-US; mso-fareast-language:DE; mso-bidi-language:EN-US; layout-grid-mode:line;}.MsoChpDefault {mso-style-type:export-only; mso-default-props:yes; font-size:11.0pt; mso-ansi-font-size:11.0pt; mso-bidi-font-size:11.0pt; font-family:"Calibri",sans-serif; mso-ascii-font-family:Calibri; mso-ascii-theme-font:minor-latin; mso-fareast-font-family:Calibri; mso-fareast-theme-font:minor-latin; mso-hansi-font-family:Calibri; mso-hansi-theme-font:minor-latin; mso-bidi-font-family:"Times New Roman"; mso-bidi-theme-font:minor-bidi; mso-fareast-language:EN-US;}.MsoPapDefault {mso-style-type:export-only; margin-bottom:8.0pt; line-height:107%;}div.WordSection1 {page:WordSection1;}ol {margin-bottom:0cm;}ul {margin-bottom:0cm;}

Round 2

Reviewer 1 Report

The authors responded appropriately to the reviewers' comments, and I think the paper has been improved.

Reviewer 2 Report

Accept in present form.